# Memory Impairments and Wellbeing in Breast Cancer Patients: A Systematic Review

**DOI:** 10.3390/jcm12226968

**Published:** 2023-11-07

**Authors:** Pedro F. S. Rodrigues, Ana Bártolo, Pedro B. Albuquerque

**Affiliations:** 1I2P—Portucalense Institute for Psychology, Portucalense University, 4200-072 Porto, Portugal; ana.bartolo@upt.pt; 2CIPsi—Psychology Research Centre, School of Psychology, University of Minho, 4710-057 Braga, Portugal; pedro.b.albuquerque@psi.uminho.pt

**Keywords:** breast cancer, woman, memory impairments, wellbeing, objective measures, subjective measures, chemotherapy

## Abstract

Breast cancer is one of the most diagnosed cancers among women. Its effects on the cognitive and wellbeing domains have been widely reported in the literature, although with inconsistent results. The central goal of this review was to identify, in women with breast cancer, the main memory impairments, as measured by objective and subjective tools and their relationship with wellbeing outcomes. The systematic literature search was conducted in the PubMed, Scopus, and ProQuest databases. The selected studies included 9 longitudinal and 10 cross-sectional studies. Although some studies included participants undergoing multimodal cancer therapies, most focused on chemotherapy’s effects (57.89%; *n* = 11). The pattern of results was mixed. However, studies suggested more consistently working memory deficits in breast cancer patients undergoing chemotherapy. In addition, some associations have been identified between objective memory outcomes (verbal memory) and wellbeing indicators, particularly depression and anxiety. The inconsistencies in the results could be justified by the heterogeneity of the research designs, objective and subjective measures, and sample characteristics. This review confirms that more empirical evidence is needed to understand memory impairments in women with breast cancer. An effort to increase the homogeneity of study methods should be made in future studies.

## 1. Introduction 

Breast cancer is one of the most common diagnoses worldwide, registering an incidence of 2,261,419 new cases in 2020 [1]. With technological and scientific advances in early diagnosis and cancer treatments [2], there is growing interest in the long-term side effects of this disease. Cognitive difficulties are recurrent symptoms reported by breast cancer patients, and they can persist for several years after the end of treatment, resulting in substantial adverse physical and psychosocial consequences [3]. Cancer-related cognitive impairment (CRCI) is a term used to describe cognitive impairment during and after cancer diagnosis and treatment [4,5]. 

Research has documented that from 17% to 75% of women diagnosed with breast cancer experience CRCI in domains related to attention, concentration, memory, and executive functions from 6 months to 20 years after chemotherapy [6,7]. Although these cognitive changes are widely considered to be a side effect of chemotherapy, sometimes referred to as “chemobrain” or “chemofog”, the body of evidence has already highlighted the potential effects of other cancer treatments (e.g., hormonal therapies, targeted therapies, and immunotherapy), pathophysiological mechanisms [8,9], and psychosocial factors [10]. In this sense, the development of CRCI seems to be multifactorial, but the identification of its causal factors has not yet been fully clarified. 

In addition, previous studies have also pointed to controversy regarding the relationship between objective cognitive problems, assessed by neuropsychological tests, and subjective cognitive problems (also known as cognitive complaints), assessed by self-report measures. Cognitive complaints in breast cancer survivors are not always associated with objective cognitive changes [11], similar to other populations, e.g., older adults [12]. According to the International Cognition and Cancer Task Force (ICCTF) [7], it is recommended that cognitive functions, such as learning, memory, processing speed, and executive functions, be assessed using objective measures. However, some authors argue that self-report measures can better detect subtle declines in cognitive functioning than neuropsychological tests [10,13]. The design heterogeneity of the studies, the lack of consistency in the cognitive functions assessed, and the measures applied could contribute to the absence of patterns and maintain uncertainty about the cognitive deficits reported by breast cancer patients [9]. 

Previous review works have presented a broad scope summarizing CRCI-related results [5,6,9,10] that more consistently point to impaired performance on memory tasks as well as memory complaints in breast cancer patients [9]. Meta-analyses have revealed the largest effects in terms of changes in memory and executive functions when compared to other cognitive domains, such as reason and perception [14]. Chemotherapy seems to be one of the therapies with the most pronounced impact, particularly on verbal working memory. Still, a more restricted evidence synthesis is needed since the current picture is inconclusive [3]. For this reason, filling a gap in the literature, this review aimed to systematize the existing evidence on specific memory deficits in breast cancer survivors (measured by objective and subjective tools), identifying guidelines for practice and future research. 

Furthermore, there has been growing interest in understanding the relationship between memory outcomes and wellbeing (usually measured by self-assessment instruments). Although it is well established that breast cancer impacts physical and psychological functioning, there is a lack of studies summarizing the relationships between memory deficits and patients’ perceived wellbeing. The limited evidence available seems to suggest a bidirectional relationship between cognitive functioning and indicators of impaired wellbeing, such as anxiety, depression, and fatigue, which can occur simultaneously as a risk factor or a consequence of potential deficits [3,5,9,10]. The current work aimed to systematize not only memory impairments in women with breast cancer but also their relationship with wellbeing indicators, specifically with psychological (depressive symptoms, anxiety, and (di)stress) and physical (physical functioning and fatigue) wellbeing. 

## 2. Materials and Methods

This review was conducted according to the Preferred Reporting Items for Systematic Reviews and Meta-analysis (PRISMA) 2020 guidelines [15]. 

### 2.1. Elegibility Criteria

The PICO framework was used to define the study inclusion criteria. Studies were included in this review if they (i) involved breast cancer patients (18–75 years old at enrollment) in the active or disease-free phase; (ii) assessed cancer-related memory deficits using subjective and/or objective measures; (iii) explored the association between memory outcomes and psychological (depressive symptoms, anxiety and (di)stress)) and physical (physical functioning and fatigue) wellbeing; (iv) were written in English, Portuguese, or Spanish; and (v) were published in a peer-reviewed journal between 2000 and 2022. Studies involving patients with brain metastases and/or a history of comorbidity with neurological and/or psychiatric conditions were excluded. Protocols, literature reviews, meta-analyses, validation studies, book chapters, commentaries, unpublished articles, and conference abstracts were also excluded. 

### 2.2. Search Strategy 

A systematic literature search was conducted in the following databases: PubMed, Scopus, and ProQuest. The first search was performed in January 2023, and then it was re-run in May 2023 to identify possible further studies. The key terms used were ‘cancer’, ‘neoplasm’, ‘tumour’, ‘carcinom’ and ‘memory’, ‘mnesic’, ‘stress’, ‘anxiety’ OR ‘depression’, ‘physical’, and ‘well-being’). The search was adapted for the 3 databases, and OR and AND functions were used to combine the above terms. Specific filters related to restrictions on publication date, language, and/or document type were applied whenever possible. 

### 2.3. Selection Process and Data Extraction 

An independent screening of the titles and abstracts obtained from the database searches was carried out by two researchers (P.F.S.R. and A.B.), and a list of studies for full-text examination was produced after removing the duplicates. This initial screening was conducted using a semiautomation tool—Rayyan (https://www.rayyan.ai/). Any disagreements were resolved through discussion with a third review author (P.B.A.). Next, the pair of raters independently extracted the full texts of the potentially eligible articles and evaluated them for inclusion. Studies whose inclusion or exclusion could not be determined with certainty based on the information in the title and abstract were also acquired for further examination. For each included study, information was gathered using a predesigned data extraction form including the following categories: (i) study characteristics such as “author”, “country”, “study design”, and “sample size”; (ii) patient characteristics namely “mean age”, “sex”, ”time since diagnosis”, and “cancer treatments”; and (ii) “assessment measures”, “main memory deficits”, and their relationship with wellbeing indicators. 

### 2.4. Quality Appraisal

Two review authors (P.F.R.S. and A.B.) independently appraised the quality of eligible studies based on the Joanna Briggs Institute (JBI) Statistics Assessment and Review Instruments critical appraisal checklists for analytical cross sectional studies and cohort studies [16,17]. Each item on these checklists was appraised as “yes”, “no”, “unclear”, or “not applicable”. Any disagreements between the revisions were resolved by discussion between all the coauthors, when necessary. The study quality was assessed based on the information available in the papers. 

## 3. Results

### 3.1. Study Selection 

A total of 1211 articles were identified from the electronic databases. After removing duplicates (*n* = 100), 1111 were screened based on title and abstract. Only 64 records were potentially eligible, and the review team retrieved and analyzed their full texts. Of these, 45 did not meet the inclusion criteria. Most of the excluded studies involved several cancer types. Two eligible articles that used the same sample in two different publications were also excluded from the results analysis [18,19]. Thus, 19 quantitative studies were included in this systematic review. The PRISMA flow diagram of the study search and selection process is shown in Figure 1. 

### 3.2. Study Characteristics 

This review included 9 longitudinal and 10 cross-sectional studies published between 2007 and 2022. The characteristics of the included studies are summarized in Table 1. About 36.84% of the studies were conducted in the United States (*n* = 7). This was also a field of interest in countries such as France (*n* = 2), the Netherlands (*n* = 1) and Germany (*n* = 1). The sample sizes included in the studies ranged from 38 [20] to 1477 [21] and involved only women diagnosed with breast cancer. The mean age of the participants ranged from 49 [22] to 62 [23] years old. About 57.89% (*n* = 11) of the studies were focused on survivors undergoing chemotherapy [20,22,24,25,26,27,28,29,30,31,32]. 

### 3.3. Memory Assessment: Objective and Subjective Measures

Neuropsychological testing provides objective assessments of memory outcomes. However, the assessment protocols used in the studies included in this review revealed high heterogeneity. Sixteen out of 19 included studies (84.2%) used cognitive test batteries. Subtests of objective measures, such as the Wechsler Adult Intelligence Scale-III (WAIS-III) and Wechsler Memory Scale (WMS-R/WMS-III/WMS-IV), were most frequently applied. The digit span was the most common test among the different protocols (*n* = 7) [24,25,29,31,35,36,38] being used to assess short-term and/or working memory. Visual memory was assessed consistently through the visual reproduction subtests of the WMS (*n* = 3) [20,23,31]. Three studies also used the Rey Auditory Verbal Learning Test to assess verbal memory [23,24,35]. Subjective memory complaints were also assessed in 42.1% of studies (n = 8) [21,25,26,27,30,33,34,38]. Only scales such as the Attentional Function Index (*n* = 2) [33,34] and the Patient’s Own Functioning Assessment Inventory (*n* = 2) [26,30] were used in more than one study. However, these are more general measures that assess other cognitive functions besides memory (Table 1 details all the objective and subjective measures used). 

### 3.4. Effect of Breast Cancer and Associated Treatments on Memory

Studies showed mixed results concerning cancer treatments*’* effects on breast cancer patients*’* memory. Six out of the nine included longitudinal studies explored the impact of chemotherapy on memory [20,25,26,28,31,33]. 

Longitudinal data obtained using objective neuropsychological measures (*n* = 3) showed that performance on tasks involving short-term memory [25] and working memory [28,33] decreased significantly after chemotherapy treatment. One record suggested that verbal working memory impairment remained even 7 months after treatment [33]. Objective outcomes regarding verbal memory were inconsistent. Two studies suggested that cancer patients undergoing chemotherapy showed no differences in verbal memory compared to a healthy control group [20], nor compared to patients who were prechemotherapy [22]. However, a longitudinal study by Vearncombe et al. [31] demonstrated worse performance on verbal memory tasks 4 weeks after administering the last course of chemotherapy. 

Regarding visual memory, the longitudinal studies (*n* = 2) did not suggest a negative impact of chemotherapy, although limited by the small sample size. Ando-Tanabe et al. [20] found no differences between the chemotherapy group and a control group, and Vearncombe et al. [31] demonstrated improvements in visual memory (e.g., visual reproduction, visual reproduction delayed, and visual recognition) from pre- to postchemotherapy. Additionally, a cross-sectional study showed that breast cancer survivors treated with chemotherapy had poorer outcomes in prospective memory (*d* = 0.80) and retrospective memory (immediate recall, *d* = 0.72; delayed recall, *d* = 0.77) compared to healthy controls [27]. In turn, Le Run and collaborators [35], focusing on cognition in breast cancer patients receiving adjuvant hormonotherapy, showed that this therapeutic did not affect performance on visual and verbal episodic memory and working memory tasks (no changes were registered during one year of follow up).

Assessment of subjective memory complaints (*n* = 8) through self-report measures also indicated that cancer patients reported more complaints after chemotherapy, which persisted even after one year (*d* = 0.15) [26]. Perceived impairment in prospective memory increased after chemotherapy treatment [25]. However, a cross-sectional study suggested that prospective and retrospective memory complaints were identical between women undergoing chemotherapy and healthy controls [27], although the results of objective measures were not in the same direction. Despite this, breast cancer survivors had more prospective than retrospective memory complaints (*d* = 1.12). A recent study also points to increased working memory complaints from the presurgery period to one month after surgery and before any adjuvant treatment [34]. 

### 3.5. Relationship between Memory Outcomes and Wellbeing Indicators

Wellbeing was measured through proxy variables, including depressive and anxiety symptoms, distress, perceived stress and worries (psychological wellbeing), physical functioning, and fatigue (physical wellbeing). Symptoms of anxiety and/or depression were most frequently assessed across studies using measures such as the Hospital Anxiety and Depression Scale [HADS] (*n* = 5) [20,21,22,31,35], State-Trait Anxiety Inventory [STAI] (n = 4) [24,32,36,38]; and Beck Depression Inventory [BDI-II] (*n* = 3) [26,37,38]. Regarding the relationship between performance on objective memory measures and depressive symptoms, the results are not conclusive. Evidence from four studies supports this association, especially in chemotherapy-treated breast cancer survivors (three out of four studies) [20,24,25]. Poorer performance on logical (*r* = −0.50; *r* = −0.77) [20], verbal (*r* = −0.76) [20], visual (*r* = −0.25) [23], and short-term memory tasks [25] was associated with higher levels of depressive symptoms in breast cancer patients. More specifically, in the study by Crouch et al. [24], depressive symptoms were a negative predictor of delayed recall in women without recurrence and undergoing chemotherapy (*β* = −0.23). However, these results should be interpreted cautiously, as five studies did not go in the same direction, suggesting that there was no association between these symptoms and performance on neuropsychological tests [29,30,31,37,38]. In addition, two studies [25,27] consistently suggested that depressive symptoms of chemotherapy patients were associated with more prospective memory complaints (*r* = 0.47) [27]. Concerning anxiety, studies involving objective cognitive tasks remain mixed. While some studies showed positive associations between performance on verbal memory tests (e.g., long-term recall) and postchemotherapy anxiety symptoms (*n* = 2) (*r* = 0.57) [20] (*r* = 0.40) [22], others showed no association with either state anxiety (*n* = 3) [30,31] or trait anxiety [38]. Nevertheless, Morel et al. [36] demonstrated that the most anxious breast patients recovered significantly less emotional details than healthy controls. Seven studies also explored the relationship between memory and physical wellbeing [21,22,23,24,25,26,27,29,30,31,32]. Three out of seven articles showed (i) a positive correlation between fatigue and performance on immediate memory tasks (*r* = 0.25) [32]; (ii) a correlation between memory domains and physical symptoms (*r* = −0.14) [37]; and (iii) an increase in fatigue levels associated with more subjective memory complaints, namely at the level of prospective (*r* = −0.55) and retrospective memory (*r* = −0.55) [27]. See more details in Table 1.

### 3.6. Study Quality Assessment

Table 2 presents the results of the critical appraisal of the studies included in this review. All studies adequately defined the sample inclusion criteria and the study context and included reliable measures. Only one of the studies did not clearly present the cognitive test battery used [28]. Around 52.63% of the studies (*n* = 10) did not identify or analyze confounding variables. Most studies involved bivariate correlation analyses (52.63%; *n* = 10) to examine the association between memory outcomes and wellbeing variables. Therefore, although the statistical analyses were appropriate in all studies, more robust statistics are needed to clarify the mixed results found. Three out of nine longitudinal studies (33.33%) did not have complete follow up and did not describe the reasons for loss to follow up [26,28,33]. Moreover, forty-four percent did not identify strategies to deal with dropouts throughout the data-collection process (*n* = 4) [28,33,34,35].

## 4. Discussion

The main goal of this systematic review was to identify memory impairments and their relationship with wellbeing indicators in women with breast cancer at the active or disease-free phase. Memory impairments included outcomes measured by objective and subjective tools, and wellbeing was assessed through indicators such as anxiety, depression, (dis)stress, and physical symptoms (e.g., fatigue). To our knowledge, this is the first systematic review to focus more narrowly on memory and to summarize its relationship with wellbeing in the context of breast cancer.

Overall, our review indicated a mixed pattern of results regarding objective memory deficits in cancer patients (see Figure 2). Most of the studies involved women who had undergone chemotherapy, and they consistently suggested impairment of short-term memory and working memory, as previously indicated in the review study by Ahles et al. [6]. A longitudinal study conducted by McDonald et al. [39] had already pointed out that frontal lobe hyperactivation for working-memory tasks decreased 1 month postchemotherapy and a return to pretreatment hyperactivation at 1-year post-treatment. Interestingly, the longitudinal studies included in this review indicated that working memory impairments were maintained over time (up to 7 months), particularly in the verbal domain. The mechanisms involved in cancer-related cognitive impairments are complex and not fully understood. However, previous studies involving breast cancer patients have demonstrated direct brain damage as a consequence of the neurotoxic effects of systematic treatments (e.g., damage to neurons or glial cells, ischemic vascular damage, and a reduced number of proliferative hippocampal cells) that may be the basis of these long-term memory complaints [40]. Evidence also showed that breast cancer survivors had increased levels of inflammation, on average, 20 years after treatment, which is a factor that impacts their cognitive functioning. Therefore, the hypothesis supported in the literature that inflammation is specifically associated with impairment of working memory may explain the results found at this level [41]. Regarding visual memory, the studies with objective measures included in this review consistently demonstrated the absence of impairment after chemotherapy treatments. However, previous meta-analyses, although limited by their broad scope, have found mixed results for both visual and verbal long-term memory (e.g., no differences: Jim et al. [42]; small to medium cumulative effect sizes: Stewart et al. [43]). These results can be explained by the significant variation in the neuropsychological measures used among studies, most of which are not in line with the guidelines of the International Cognition and Cancer Task Force [7]. Furthermore, the possible compensatory activation of other brain areas during cognitive tests in a controlled environment may cover up more subtle deficits, as previously suggested by imaging studies [44].

The results of subjective memory complaints are limited by the small number of studies and the lack of specificity of the measures for assessing this cognitive domain. There is some preliminary evidence that breast cancer survivors reported more subjective memory complaints (e.g., long-term verbal memory and working memory) after chemotherapy [25,26,30] and surgery [34]. However, there is still inconsistent data. The literature has suggested that, in general, cancer survivors report cognitive problems but score in the normal range on neuropsychological tests [9]. Our study is not entirely in line with this finding. For example, prospective memory impairments are found in objective measures, which are not visible in self-report questionnaires [27]. The sensitivity of the subjective measures used may also be at the root of this finding. Inconsistencies in memory complaints occur in cancer survivors [45,46] and other populations (e.g., older adults [13]). However, this review suggests the possible correspondence between objective and subjective results regarding impairments in verbal memory and working memory, reinforcing the need to further explore these domains in women with breast cancer. The review study by Bray et al. [11] had already reported marginally significant results regarding the association between self-report measures and neuropsychological tests involving memory assessment. In other contexts (e.g., senior context without disease), for example, subjective memory complaints, have been shown to be predictors of objective declines [47], which can also be analyzed in future studies.

In addition, evidence from this review supports a relationship between memory outcomes and indicators of wellbeing. Cognitive complaints have generally been associated with emotional factors (e.g., depression and anxiety [27]) and fatigue [11], which may justify disparities in results, but not with objective measures. Although the findings of the included studies were inconsistent, some significant associations were found between objective memory impairments and depression, anxiety, and fatigue. Impairment in verbal memory tasks was more frequently associated with psychological distress [20,22,24]. Our results also confirm the significant association between self-reported prospective memory impairments and depressive symptoms and fatigue in women undergoing chemotherapy [25,27]. However, uncertainty remains as to whether memory complaints are related to brain dysfunction due to the treatments, which can lead to affective symptoms, or whether they are a consequence of mood disorder and fatigue [11]. While fatigue has been relatively underexplored in the included studies, the literature has indicated that many cancer patients experience long-term fatigue [8]. Chronic fatigue emerges as a recurring comorbidity in cancer and may lead to memory and cognitive impairments [48]. This close connection presents challenges in pinpointing the origin of the cognitive deficits experienced, potentially contributing to the variability found in this review. Other comorbidities or the severity of treatment-related side effects themselves may be linked to the observed memory deficits and could help elucidate some of the highlighted inconsistencies. Therefore, more studies should be conducted to help clarify these complex relationships, namely by understanding the underlying mechanisms.

The results of the current review should be interpreted with caution for several reasons. First, this work included studies with heterogeneity of designs (i.e., longitudinal vs. cross-sectional studies) and lack of consistency of cognitive measures applied; additionally, study-design characteristics (i.e., type of control group, timing of assessments), and patient characteristics (i.e., age, education, and time since chemotherapy) seem to be heterogeneous in the selected studies. Also, the included studies incorporated several types of therapeutics, although the most predominant was chemotherapy; finally, around half of the studies did not identify or control for confounding variables, which limits the conclusions drawn.

In future empirical studies, it would be crucial to involve breast cancer patients (i) with a similar time of diagnosis and (ii) other therapeutics, namely hormonotherapy, which has been associated with impairments in various cognitive domains (e.g., attention and executive functions [9]). The specificities of the treatments administered may be decisive in explaining the variability found in this study, especially considering the recent evolution of cancer treatments. For example, targeted therapies with the anti-HER-2 drug trastuzumab emtansine appear to be associated with less cognitive impairment compared to regimens involving chemotherapy [49]. Nevertheless, the evidence is still evolving, and further studies are required with a specific focus on comparing various therapeutic approaches regarding their impact on specific cognitive functions. Moreover, standardized instruments should be used to measure memory and wellbeing indicators; indeed, we can find diverse instruments in the included studies, mainly to measure memory. Longitudinal studies should be preferred to cross-sectional studies, as these designs may provide more rigorous evidence [6,50]. It is also important to reinforce the conduct of research in which a baseline of memory is measured at the time of diagnosis (and before any treatment) to compare with memory assessed after treatments. The assumption that patients do not have intact cognitive abilities before treatment may contribute to inconsistent and inconclusive results in this context. Indeed, the International Cognition and Cancer Task Force recommends guidelines to increase the homogeneity of study methods and suggests that “Cross-sectional, post-treatment only studies with appropriate comparison groups might be useful for exploratory analysis, hypothesis-generating, and for proof-of-concept trials, with findings confirmed longitudinally” [7] (p. 704).

## 5. Conclusions

This work provides an important review of (objective and subjective) memory impairments and their relationship with wellbeing indicators in breast cancer survivors. Although a mixed pattern of results has been found, this review highlights some declines in memory, measured by objective and subjective instruments, as well as some associations with wellbeing indicators such as anxiety and depression. This review highlights the need for additional empirical evidence regarding memory decline in the context of breast cancer. Achieving this necessitates the standardization of methodological procedures and measures used, as well as a deeper understanding of the effects of various therapeutic approaches, particularly the underexplored realm of targeted therapies. This review highlights the need for additional empirical evidence regarding memory decline in the context of breast cancer. Achieving this necessitates the standardization of methodological procedures and measures used, as well as a deeper understanding of the effects of various therapeutic approaches, particularly the underexplored realm of targeted therapies. With this study, our expectation is to provide a reference to assist researchers and professionals in identifying significant memory deficits that should be considered in the assessment of cancer patients, as well as identify gaps in the evidence that should receive greater attention from the scientific community.

## Figures and Tables

**Figure 1 jcm-12-06968-f001:**
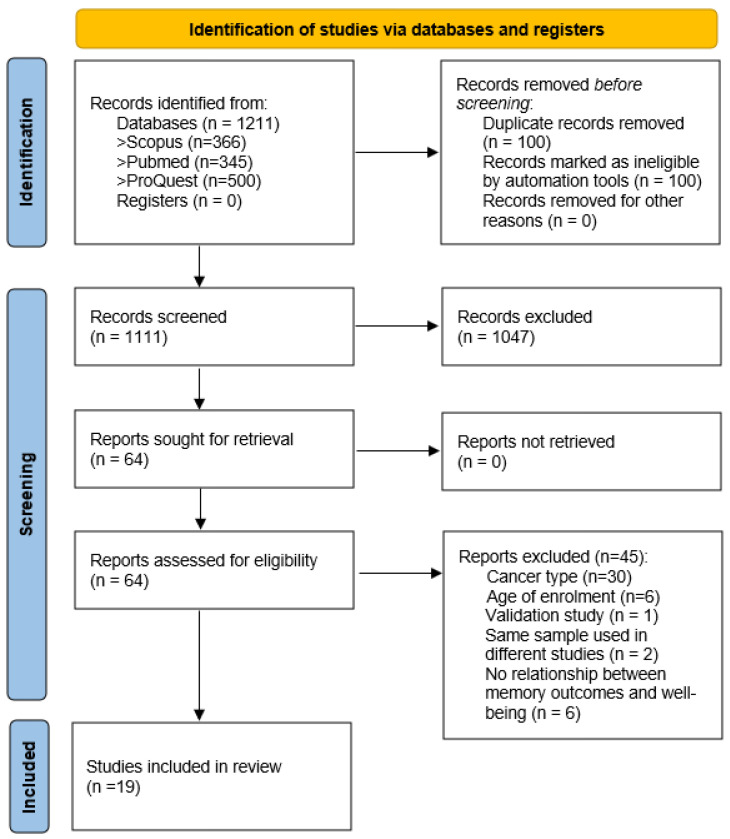
PRISMA flow diagram of the selection process.

**Figure 2 jcm-12-06968-f002:**
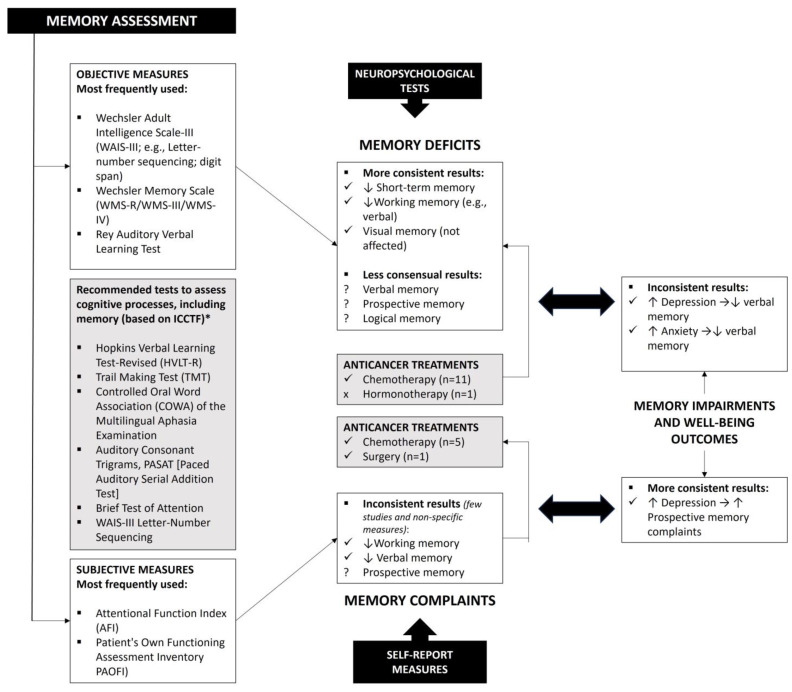
Summary of the main results and guidelines for future studies. * Note: International Cognition and Cancer Task Force.

**Table 1 jcm-12-06968-t001:** Characteristics and results of the included studies.

First Author Name, Year	Country	Study Design	Sample Size (N)	Sample Characteristics	Mean Age [M(SD)]	Memory Measures	Wellbeing Indicators	Data Analysis	Main Results
Ando-Tanabe et al., 2014 [20]	Japan	Longitudinal study (T1: before; and T2: 1 month after chemotherapy).	18 breast cancer vs. 20 controls.	Chemotherapy-treated breast cancer survivors.	Cancer group: 51.2 (10.7); Controls: 60.2 (11.4).	Logical memory I and II from the Wechsler Memory Scale-Revised (WMS-R); Verbal paired associates I and II from the WMS-R; Visual reproduction I and II from the WMS-R.	Hospital Anxiety and Depression Scale (HADS).	Independent *t*-test; Pearson correlations.	No group differences were found for the memory performance tasks. In the chemotherapy group, ↓ logical memory I and II (*r* = −0.77, *r* = −0.50, *p* < 0.05, respectively) and ↓verbal memory (containing logical memory I; *r* = −0.76, *p* < 0.001) were significantly correlated with ↑ depressive symptoms; and ↓ verbal paired associates II were associated with anxiety (*r* = 0.57; *p* < 0.05).
Boele et al., 2015 [23]	Netherlands	Cross-sectional study	107 (20 adjuvant tamoxifen vs. 43 surgical operation/radiotherapy vs. 44 healthy control).	Women who had undergone breast surgery with or without radiotherapy and adjuvant tamoxifen (AT group), and women who had experienced only breast surgery with or without radiotherapy (SR group).	AT: 61.60 (6.14) SR: 62.16 (7.99) Controls: 62.02 (6.32).	Rey auditory verbal learning test. Visual association test. Visual memory subtest (WMS; immediate recall, delayed recall). Letter-number sequencing (WAIS-III).	The European Organization for Research and Treatment of Cancer Quality of Life Questionnaire Core 30 (QLQ-C30): physical functioning and fatigue subscales. The 25-item Hopkins. Symptom Checklist: depression and anxiety subscales.	Pearson’s Correlations and Multivariate analysis of variance.	ANOVAs revealed statistical differences among three groups in fatigue, specifically, AT presented the highest fatigue, the SR intermediate fatigue and HC the lowest fatigue. A negative correlation was found between depression and visual memory (*r* = −0.249, *p* = 0.010) for the three groups (combined). No other significant correlations were found between memory domains and the self-reported measures (*p* > 0.005).
Crouch et al., 2022 [24]	USA	Cross-sectional study	335 Breast Cancer Survivors	Breast Cancer Survivors, 3–8 years postdiagnosis for stage I–IIIA breast cancer without recurrence and treated with adjuvant chemotherapy.	63.9	Rey Auditory-Verbal Learning Test (AVLT) Digit Span Backward.	Center for Epidemiologic Studies Depression Scale (CES-D) Spielberger State-Trait Anxiety Inventory (STAI)-State Functional Assessment of Cancer Therapy-Fatigue (FACT-F).	Linear regression models	Depressive symptoms were negatively related to delayed recall; the model accounted for 3% of the variance (*β* = −0.23, *p* < 0.01). More depressive symptoms were negatively related to delayed recall.
Hsu et al., 2021 [22]	China	Cross-sectional study	45 prechemotherapy vs. 30 postchemotherapy [3–9 months]) vs. 30 controls.	Chemotherapy-treated breast cancer survivors.	Prechemotherapy: 51.29 (11.25); Postchemotherapy: 48.60 (9.98); Controls: 47.50 (10.88).	Word List subtest of the Taiwanese version of the Wechsler Memory Scale–Third Edition (WMS-III).	Hospital Anxiety and Depression Scale (HADS-A); Patient Health Questionnaire (PHQ-9); Brief Fatigue Inventory (BFI).	Analysis of covariance (ANCOVA); Pearson correlations.	There were no differences between the groups regarding performance on the memory task. A positive association was found between long-delay recall and anxiety levels only in the postchemotherapy group (*r* = 0.40, *p* < 0.05).
Huang et al., 2019 [25]	China [Anhui]	Case-control study (T1:before; and T2: after chemotherapy).	63 (29 patients with depression; 34 patients without depressive symptoms).	Breast cancer survivors receiving chemotherapy with paclitaxel and doxorubicin.	Depression group: 50.66 (7.76) Nondepression group: 47.35 (8.56)	Neuropsychological background tests (e.g., mini-mental state examination [MMSE]; digit span test (DS); verbal fluency test [VFT]—short-term memory; Prospective memory questionnaire [PMQ].	Depression [Self-rating depression scale—SDS].	Paired-sample *t*-tests; 2 independent samples *t*-tests.	Breast cancer survivors had lower MMSE scores (short-term memory) at time T2 compared to T1 (*p* < 0.05); Depressed cancer patients after chemotherapy showed ↓ MMSE scores (short-term memory) and ↓ verbal fluency (*p* < 0.05). Data suggested ↑ prospective memory impairment after chemotherapy (*p* < 0.001); ↑ prospective memory impairment in the depression group after chemotherapy compared to the nondepression group (*p* < 0.001).
Jung et al., 2017 [33]	United States	Longitudinal study (T1: before adjuvant therapy; T2: 1 month after chemotherapy; and T3: 7 months after chemotherapy).	36 breast cancer patients awaiting adjuvant chemotherapy vs. 41 awaiting radiotherapy without chemotherapy vs. 39 controls.	Surgically-treated breast cancer survivors.	Chemotherapy: 49.68 (9.74); Nonchemotherapy: 53.94 (8.42) and Controls: 51.13 (8.47).	Attentional Function Index (AFI); Verbal Working Memory Task (VWMT) during fMRI scanning.	Breast Cancer Prevention Trial Symptoms Scale; Three-Item Worry Index.	*t*-tests, Analysis of variance (ANOVA) and multivariable regression models.	There were no changes in verbal working memory over time for the cancer group in chemotherapy. The chemotherapy-treated cancer group showed ↓ performance on the verbal working memory task compared to controls (*p* < 0.05) even 7 months after treatment. In the model predicting memory deficits, being in the ‘chemotherapy group’ was significantly associated with worse performance at T3 (*B* = 0.79, *p* = 0.007). No group effects were found for self-reported cognitive complaints (including working memory). However, ↑ worry (*B* = −0.32, *p* = 0.013) and ↑ distress symptoms (*B* = −1.36, *p* < 0.001) at T3 were significant predictors of complaints.
Jung et al., 2020 [34]	South Korea	Pre-postdesign (T1: week before any planned surgery; T2: 1 month following baseline assessment).	132	Breast cancer patients before any treatment.	50.80 (9.96).	Attentional Function Index (AFI).	>Functional Assessment of Cancer Therapy-General (FACT-G); >Patient Health Questionnaire (PHQ); >Pittsburgh Sleep Quality Index (PSQI).	Pearson correlations	Memory function ↓ from presurgery to 1-month postsurgery (*p* < 0.05); ↓ lower postsurgery health-related quality of life associated with ↓ memory function (*r* = 0.28, *p* = 0.001)
Le Rhun et al., 2015 [35]	France	Longitudinal study (M0: Baseline before any hormonal treatment; M1: 6 months; M2: 1 year after treatment).	74 (Tamoxifen—37; Aromatase inhibitor—37)	Breast cancer survivors in tamoxifen treatment vs aromatase. inhibitor treatment.	Mdn = 62.0 (tamoxifen group) vs. Mdn = 61.0 (Aromatase group).	Rey auditory verbal learning test (RAVLT). Rey auditory Benton Visual Retention Test (BVRT). Forward and backward digit span. Forward spatial span.	The Hospital Anxiety and Depression Scale (HADS).	Analysis of covariance (ANCOVA). Mixed model. analyses of variance (Mixed ANOVAs).	Considering memory measures, there were no differences between the groups during the 6-month and 1-year follow up. The pattern of results remained similar after controlling HADS scores (*p* > 0.12).
Merriman et al., 2017 [26]	USA	Longitudinal study (Before systemic therapy vs. 6 months vs. 12 months vs. 18 months of follow up).	368 (158: Aromatase inhibitor alone; 104: chemotherapy followed by aromatase; 106: controls).	Women newly diagnosed with stage I-IIIA breast cancer; who completed surgery; and were scheduled to receive anastrozole alone or chemotherapy followed by anastrozole.	61.7 (6.42): aromatase inhibitor alone vs. 59.4 (5.49): chemotherapy followed by aromatase vs. 58.7 (5.91): control group.	Patient Assessment of Own Functioning Inventory (PAOFI): subscale memory.	Beck Depression Inventory-II; POMS-fatigue/inertia subscale.	Analysis of variance. Multilevel regressions.	Patients who received chemotherapy reported poorer memory (*p* < 0.001, *d* = 0.15), from before to after chemotherapy. These changes persisted after one year of anastrozole for memory (*p* = 0.005, *d* = 0.18). Patients who received chemotherapy reported poorer memory than the women who received anastrozole alone (*p* = 0.006, *d* = 0.13).
Morel et al., 2015 [36]	France	Cross-sectional study	31 breast cancer patients vs. 49 controls.	Breast cancer patients who had not yet undergone chemotherapy.	Cancer group: 53.6 (5.2); Controls: 54.2(6.6).	Two tests of verbal and visual episodic memory processes, based on the Encoding, Storage, Retrieval (ESR) paradigm; Digit Span Backward, Letter–Number Sequencing, and Arithmetic subtests of the Wechsler Adult Intelligence Scale (WAIS), Trail Making Test (TMT) Parts A and B, formal and semantic verbal fluency, and d2 Test of Attention.	State-Trait Anxiety Inventory (STAI).	Factorial ANOVA.	Most anxious breast patients retrieved significantly fewer emotional details than the controls (*p* = 0.01).
Paquet et al., 2018 [27]	Canada	Cross-sectional study	80 breast cancer survivors vs. 80 controls.	Chemotherapy-treated breast cancer survivors.	Breast cancer survivors: 54.1 (9.5); Controls: 54 (9.4): (30–75 years).	Prospective and Retrospective Memory Questionnaire (PRMQ); Memory for Intention Screening Test (MIST)—prospective memory; Standardized Logical Memory Test from the Wechsler Memory Scale-IV—retrospective memory.	>Functional Assessment of Cancer Therapy: Fatigue subscale; >20-item Center for Epidemiologic Studies Depression Scale.	Analysis of covariance (ANCOVA); Pearson correlations.	There were no group differences related to memory complaints; Cancer patients reported more prospective than retrospective memory complaints (*p* < 0.001; d = 1.12). Cancer group: ↑ fatigue was associated with ↑ prospective memory complaints (*r* = −0.547; *p* < 0.01) and ↑ retrospective memory complaints (*r* = −0.545; *p* < 0.01); ↑ depression symptoms were also related to increased memory complaints (*r* = 0.467, *p* < 0.01 and *r* = 0.475, *p* < 0.01, respectively prospective and retrospective);: Breast cancer survivors presented ↓ prospective memory functioning than controls (*p* < 0.001; *d* = 0.8) and ↓ retrospective memory (e.g., ↓ immediate recall [*p* < 0.001; *d* = 0.72] and ↓ delayed recall [*p* < 0.001; d = 0.77). Cancer group: No significant associations were found between performance on memory tasks and depression and fatigue.
Phillips et al., 2017 [21]	United States	Longitudinal (baseline and 6-month follow up).	1477	Post-treatment breast cancer survivors.	56.3 (9.3)	Frequency of Forgetting Questionnaire.	Hospital Anxiety and Depression Scale; Perceived Stress Scale; Concerns About Recurrence Scale.	Panel analyses	↑ levels of distress (*β* = −0.31) and fatigue (*β* = −0.18) were associated ↑ subjective memory impairment. This result was maintained at 6 months of follow up.
Shilling et al., 2007 [28]	United Kingdom	Longitudinal study (T1: baseline; T2: 4 weeks after the final chemotherapy session; and T3: 12 months after the final chemotherapy session).	142 (126 completed cognitive assessment at T3).	Women receiving adjuvant therapy for breast cancer.	Chemotherapy: 51.71 (9.41); Nonchemotherapy: 59.43 (7.03).	Cognitive test battery (measures covering the functional areas of verbal and visual memory with both immediate and delayed recall, working memory, processing speed, vigilance and executive function).	General Health Questionnaire (GHQ12); Functional Assessment of Cancer Therapy questionnaire (Breast) (FACT B); and the fatigue (F); and endocrine symptoms (ES).	Independent *t*-tests; Odds ratios (Ors).	Women in the chemotherapy group were significantly more likely to report memory problems (OR 5.01, 95% CI 2.31–10.90, *p* < 0.001). At T2, a strong positive association between reporting feeling down and/or worried and reporting memory problems was found (OR 5.41, 95% CI 2.44–11.99, *p* < 0.001).
Small et al., 2019 [29]	USA	Cross-sectional study	47 Breast Cancer Survivors.	Breast cancer survivors who had been treated for Stage I-II breast cancer. with a minimum of four cycles of chemotherapy.	53.3 (6.5)	Dot Memory Hopkins Verbal Learning Test Digit Span.	Centers for Epidemiologic-Depression Scale (CEDS). Fatigue Symptom Inventory (FSI; 16).	Multilevel models.	Between-person differences in average levels of fatigue in daily life, as well as depressed mood, were unrelated to memory performance (*p* > 0.05).
Van Dyk et al., 2018 [37]	USA	Cross-sectional study	189 (28—no adjuvant, 64—Radiotherapy only, 20 -Chemotherapy only, 77—Chemo + Rad).	Patients who had a recent early-stage breast cancer diagnosis had completed primary treatment within the last 3 months.	No adjuvant: 51.75 (6.08) vs. Rad only: 53.88 (7.95) vs. 4 chemo only: 6.95 (8.06) vs. Chemo + rad 50.31 (8.88).	CVLT-II List A Long Delay Free Recall; WMS-III LM II; BVMT-R Delayed Recall; ROCFT 3 min Delayed Recall.	Multidimensional Fatigue Symptom Inventory–Short Form (MFSI) Physical and the Breast Cancer Prevention Trial Symptom Checklist (BCPT); Beck Depression Inventory (BDI-II).	Correlations	There are minimal treatment-related neuropsychological differences in neuropsychological measures in early breast cancer survivorship. Specifically, the memory domain correlated negatively with BCPT (*r* = −0.16, *p* = 0.004), and with physical symptoms (*r* = −0.14, *p* = 0.006). BDI-II was not a predictor of memory performance.
Vardy et al., 2017 [30]	Canada	Cross-sectional study	CTh + CS + N = 44; CTh + CS − N = 52; CTh − N = 30 (CTh—Chemotherapy; CS—Cognitive symptoms).	Women receiving adjuvant or neoadjuvant chemotherapy for breast cancer vs. women not receiving chemotherapy.	Mdn = 48.39 (CTh + CS +); Mdn = 48.39 (CTh + CS −); Mdn = 54.10 (CTh −).	Cambridge Neuropsychological Test Automated Battery (CANTAB). Clinical neuropsychological tests 34-item Patient’s Assessment of Own Functioning Inventory (PAFI).	FACT-F fatigue subscale. The 12-item General Health Questionnaire (anxiety and depression).	Spearman correlations. ANOVAs.	There was a weak association between CANTAB GDS and the PAFI cognitive domain (*rho =* −0.27, *p* = 0.005). Patients who did not receive chemotherapy (CTh-) scored lower on verbal learning and memory (*p* = 0.054). A worse memory was reported by CTh + CS +N (27.2), followed by CTh + CS − N (38.0), and followed by CTh − N (38.4), *p* < 0.001 (lower values correspond to more symptoms).
Vearncombe et al., 2009 [31]	Australia	Longitudinal study (T1: after surgery but before the commencement of chemotherapy; T2—approximately 4 weeks after administration of the last course of chemotherapy).	159 (138 Breast cancer survivors scheduled to receive standard-dose adjuvant chemotherapy + 21 Breast cancer survivors scheduled to receive no chemotherapy).	Breast cancer survivors scheduled to receive standard-dose adjuvant chemotherapy vs. breast cancer survivors scheduled to receive no chemotherapy).	49.38 (7.92): with chemotherapy vs. 53.98 (8.24): without chemotherapy.	Auditory Verbal Learning Test (AVLT). WMS-III-Visual Reproduction Immediate. WMS-III Visual Reproduction Delayed. WMS-III Visual Reproduction Recognition. WAIS-III—Backward Digit Span.	Functional Assessment of Chronic Illness Therapy—fatigue scale Hospital Anxiety and Depression Scale (HADS).	Multiple binary logistic regressions correlations.	Decline in working memory (digit span) performance was associated with poorer initial emotional functioning (*r* = 0.21; *p* < 0.02).Pearson correlations between change in cognitive measures (T2-T1) and health and psychological measures did not reveal significant associations with memory performance. In the group with chemotherapy, results revealed (T2 – T1) a decrease in verbal memory (*p* < 0.001) and an increase in visual memory (*p* < 0.001)
Von Ah et al., 2015 [32]	USA	Cross-sectional study	88	Breast cancer survivors, postmenopausal, underwent at least 12 months of postcancer treatment, including chemotherapy.	56.7 (8.54)	Rey Auditory Verbal Learning Test (AVLT), Rivermead Behavioral Memory Test (RBMT).	Center for Epidemiologic Studies Depression Scale (CES-D). The Functional Assessment of Cancer Therapy-Fatigue (FACT-F). State-Trait Anxiety Inventory-State Subscale (STAI-S).	Pearson’s correlation coefficient.	There was a positive correlation between fatigue and immediate memory (AVLT), *r* = 0.25, *p* < 0.05).
Wirkner et al., 2017 [38]	German	Cross-sectional study	51 (20 Breast Cancer Survivors Group and 31 Healthy Control Group).	Breast cancer survivors have undergone medical treatment including surgery, chemotherapy, and endocrine therapy, no longer than 7 years ago.	52.75 (BCS) and 51.74 (Control)	Subtests digit span forward and backward (Wechsler Memory Scale revised, WMS-R). Logical memory I and II of the WMS-R). Verbal memory (Verbaler Lern- und Merkfähigkeitstest, VLMT) A recognition memory test in which 90 old pictures Memory performance on an analogical scale from 0 (very bad) to 100 (very good).	Trait version of the State-Trait Anxiety Inventory. Beck Depression Inventory (BDI-II. Multidimensional Fatigue Inventory. Fragebogen erlebter Defizite der Aufmerksamkeit, FEDA (fatigue).	Correlations	Depression and trait anxiety were not related to neuropsychological test performance in BCS. BCS showed poorer performance in verbal memory tasks (VLMT and the WMS-R logical memory) compared to the control group. Poorer performance was also found in the digit span (forward) subscale.

**Table 2 jcm-12-06968-t002:** Critical appraisal of the included studies [16,17].

								Longitudinal studies			
First author name, year	Were the criteria for inclusion in the sample clearly defined?	Were the study subjects and the setting described in detail?	Was the exposure measured in a valid and reliable way?	Were objective, standard criteria used for measurement of the condition?	Were confounding factors identified?	Were strategies to deal with confounding factors stated?	Were the outcomes measured in a valid and reliable way?	Was appropriate statistical analysis used?	Was the follow up time reported and sufficient to be long enough for outcomes to occur?	Was the follow up complete, and if not, were the reasons for loss to follow up described and explored?	Were strategies to address incomplete follow up utilized?
Ando-Tanabe et al., 2014 [20]	Yes	Yes	Yes	Yes	No	No	Yes	Yes	No	Yes	NA
Boele et al., 2014 [23]	Yes	Yes	Yes	Yes	No	No	Yes	Yes	NA	NA	NA
Crouch et al., 2022 [24]	Yes	Yes	Yes	Yes	No	No	Yes	Yes	NA	NA	NA
Hsu et al., 2021 [22]	Yes	Yes	Yes	Yes	Yes	Yes	Yes	Yes	NA	NA	NA
Huang et al., 2019 [25]	Yes	Yes	Yes	Yes	No	No	Yes	Yes	Unclear	Yes	NA
Jung et al., 2017 [33]	Yes	Yes	Yes	Yes	Yes	Yes	Yes	Yes	Yes	No	No
Jung et al., 2020 [34]	Yes	Yes	Yes	Yes	Yes	Yes	Yes	Yes	No	Yes	No
Le Rhun et al., 2015 [35]	Yes	Yes	Yes	Yes	Yes	Yes	Yes	Yes	Yes	Yes	No
Merriman et al., 2017 [26]	Yes	Yes	Yes	Yes	Yes	Yes	Yes	Yes	Yes	No	Unclear
Morel et al., 2015 [36]	Yes	Yes	Yes	Yes	No	No	Yes	Yes	NA	NA	NA
Paquet et al., 2018 [27]	Yes	Yes	Yes	Yes	Yes	Yes	Yes	Yes	NA	NA	NA
Phillips et al., 2017 [21]	Yes	Yes	Yes	Yes	Yes	Yes	Yes	Yes	Yes	Yes	Yes
Shilling et al., 2007 [28]	Yes	Yes	Yes	Yes	No	No	Yes	Yes	Yes	No	No
Small et al., 2019 [29]	Yes	Yes	Yes	Yes	No	No	Yes	Yes	NA	NA	NA
Van Dyk et al., 2018 [37]	Yes	Yes	Yes	Yes	Yes	Yes	Yes	Yes	NA	NA	NA
Vardy et al., 2017 [30]	Yes	Yes	Yes	Yes	No	No	Yes	Yes	NA	NA	NA
Vearn-combe et al., 2009 [31]	Yes	Yes	Yes	Yes	No	No	Yes	Yes	Yes	Yes	NA
Von Ah et al., 2015 [32]	Yes	Yes	Yes	Yes	Yes	Yes	Yes	Yes	NA	NA	NA
Wirkner et al., 2017 [38]	Yes	Yes	Yes	Yes	No	No	Yes	Yes	NA	NA	NA

## Data Availability

Not applicable.

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
