# Peer review of "Memory Impairments and Wellbeing in Breast Cancer Patients: A Systematic Review"

_jcm, 2023, doi:10.3390/jcm12226968_

Round 1
Reviewer 1 Report
Comments and Suggestions for Authors
Firstly, congratulations to the authors for their exceptional work. The subject matter, the methodology used and the results obtained are all accurate and make a significant contribution to the ever-growing field of chemo-induced cognitive impairment. This is an area that is gaining increasing attention both in terms of epidemiology and intervention, due to the rising survival rates of cancer patients which is leading to the exploration of new perspectives for managing symptoms and improving patients' overall quality of life.
Author Response
Dear Reviewer,
Thank you so much for your comment. We are greatly pleased by your acknowledgment of the significance of the work undertaken.
Reviewer 2 Report
Comments and Suggestions for Authors
Overall comments:
The authors should be applauded for trying to learn more about this subject of the benefit of breast cancer patients. But this systematic review was not well executed.
The title as well as the study question is too broad, in this reviewer’s opinion, and in another sense, inappropriately specific. The rationale for memory impairments specifically is not well explained. Many of the measures described in these studies were more about executive function and other facets of cognitive impairment, other than memory. The decision to include studies with patients both receiving treatment and post-treatment also potentially confounded results.
It seems the authors kept their search criteria broad to maximize the sample of studies, but the search strategy was inadequate to find the right collection of studies. PubMed and Scopus are fine, but in some cases are redundant. Adding more search engines, like CINHAL, would have included studies from the nursing and gerontological literature which would have been relevant to this research topic. It is unclear if the team includes a librarian, but such a collaborator would be helpful in designing a more robust system review strategy.
The inconsistent results identified were partly a result of the unclear eligibility criteria. Perhaps the authors can revisit their main questions to answer with a systematic review, revisit their criteria and reconstitute their team and try again. It is a good idea, but just not well executed.
Author Response
Dear Reviewer,
We appreciate the thoughtful observation and reflection of the reviewer. Nevertheless, we would like to elucidate our perspective and decisions on certain points.
1) Concerning electronic databases, we conducted our search using the databases accessible to us at the institutional level. CINHAL is not accessible to us. However, we attempted to compensate for this limitation by utilizing the PROQUEST database, which encompasses a wide range of studies in the field of nursing.
2) Several other previously published reviews in the Journal of Clinical Medicine have also employed the same databases, specifically Scopus and PubMed.
Reference [Example]:
Sharbafshaaer, M.; Gigi, I.; Lavorgna, L.; Esposito, S.; Bonavita, S.; Tedeschi, G.; Esposito, F.; Trojsi, F. Repetitive Transcranial Magnetic Stimulation (rTMS) in Mild Cognitive Impairment: Effects on Cognitive Functions—A Systematic Review. J. Clin. Med. 2023, 12, 6190. https://doi.org/10.3390/jcm12196190
Additionally, as mentioned in the article, we utilized an automated screening tool (Ryyan), and it identified only 100 duplicate studies across all the databases.
3) Considering that systematic reviews can focus on broad or narrow questions (Thomas et al., 2023), our review provides an overview of memory deficits in cancer survivors and their association with well-being. However, we believe it maintains a rigorous approach consistent with systematic review methodology. We were careful to establish criteria in alignment with the PICO framework. We precisely defined the context, population, variables and relationships that we were going to analyze in our study. In addition, reviews can be broader or more specific in their focus, as long as they are oriented towards the starting question.
Reference [Example]:
Thomas J, Kneale D, McKenzie JE, Brennan SE, Bhaumik S. Chapter 2: Determining the scope of the review and the questions it will address. In: Higgins JPT, Thomas J, Chandler J, Cumpston M, Li T, Page MJ, Welch VA (editors). Cochrane Handbook for Systematic Reviews of Interventions version 6.4 (updated August 2023). Cochrane, 2023. Available from www.training.cochrane.org/handbook.
4) Furthermore, the inconsistent results are not the result of the criteria defined but rather of the evidence that has, in fact, showed inconsistent.
Reference [Example]:
McDougall GJ Jr, Oliver JS, Scogin F. Memory and cancer: a review of the literature. Arch Psychiatr Nurs. 2014;28(3):180-186. doi:10.1016/j.apnu.2013.12.005
5) We also fully agree that there is a great disparity of measures for assessing memory, some of which are more directly related to the assessment of executive functions, and we criticize this in the discussion [please see pages 18 and 19]. However, these are the measures used by the authors and concretely defined in the articles as primary tools for memory assessment. This review also allows us to identify these gaps and reflect on the path that needs to be taken towards a more rigorous assessment of cancer-related memory impairment.
Reviewer 3 Report
Comments and Suggestions for Authors
Partial cognitive impairment due to chemotherapy for cancer treatment is a well-known fact that is supported by ample clinical data. However, systematic reviews of therapeutic treatments in different demographics are always needed to obtain an updated database with all possible variabilities. Under this context, the current work is important. The authors have put an extensive background of the work with well-described data and discussion.
1. Whatever the mode of administration of the therapy, we know this treatment has an important impact on anxiety, cognitive function, depression, and pain ...The best option would have been to perform the baseline evaluation before administration of any kind of treatment.
2. Chronic fatigue is one of the main side effects accompanying the described cognitive impairments known as 'chemo-brain' I would have wished to have learned a little bit more about this very important side effect that surely impacts cognitive function and how the two are linked to each other
3. The side effects should be correlated with the severity of treatment outcome and quality of life. I would have wished to see a correlation and appropriate stats to investigate this. Does the good outweigh the evil?
4. It would be beneficial to include specific quantitative data or statistics of studies to provide a more concrete understanding of the outcomes also in the results text.
5. Go beyond summarizing the results and offer interpretations. For example, discuss why certain memory domains, like short-term and working memory, may be more affected in breast cancer patients. Consider integrating findings from other studies in the discussion to support your interpretations.
6. the evolving landscape of breast cancer treatment, especially with the introduction of targeted therapies like CDK inhibitors and trastuzumab emtansine, raises an intriguing question about potential impacts on cancer-related cognitive impairment (CRCI). Consider including a section or discussion on their potential impact on cancer-related cognitive impairment (CRCI) compared the chemotherapy. Overall, the effect of treatment type should be discussed more broadly.
7. Discuss the practical implications of your findings for breast cancer patients, clinicians, and researchers. How can these findings inform patient care, treatment decisions, or interventions aimed at addressing memory impairments in this population?
8. The research methods chosen for this study are adequate, the results are well-presented and discussed, and the work lies within a topic of high current interest.
Minor editing of English language required
Author Response
Dear Reviewer,
We would like to express our gratitude for providing us with constructive comments, resulting in, what we believe, an improved MS.
Below you will find your comments and our replies.
******
Partial cognitive impairment due to chemotherapy for cancer treatment is a well-known fact that is supported by ample clinical data. However, systematic reviews of therapeutic treatments in different demographics are always needed to obtain an updated database with all possible variabilities. Under this context, the current work is important. The authors have put an extensive background of the work with well-described data and discussion.
A: Thank you for your valuable comment.
- Whatever the mode of administration of the therapy, we know this treatment has an important impact on anxiety, cognitive function, depression, and pain ...The best option would have been to perform the baseline evaluation before administration of any kind of treatment.
A: Thank you for your observation. We have included a comment on it in the discussion [please see page 18, lines 398-402].
- Chronic fatigue is one of the main side effects accompanying the described cognitive impairments known as 'chemo-brain' I would have wished to have learned a little bit more about this very important side effect that surely impacts cognitive function and how the two are linked to each other.
A: We tried to delve a little deeper into this issue in the discussion, since the evidence available at this level was still limited. In fact, there are several studies that link chronic fatigue and a broader concept of cognitive impairments, but the same is not true when we specify memory [please see pages 17-18, lines 362-371].
- The side effects should be correlated with the severity of treatment outcome and quality of life. I would have wished to see a correlation and appropriate stats to investigate this. Does the good outweigh the evil?
A: This is an excellent observation. However, we decided to concentrate our studies on analyzing the relationship between memory deficits and well-being. Most of the correlations were exploratory, and other side effects associated with treatments were not reported. Future studies may take this impact into consideration, as we discussed in our paper [please see page 17, lines 368-371].
- It would be beneficial to include specific quantitative data or statistics of studies to provide a more concrete understanding of the outcomes also in the results text.
A: Thank you very much for your observation. We took this into consideration and highlighted some data in text when studies reported this information [please see pages 13 and 14].
- Go beyond summarizing the results and offer interpretations. For example, discuss why certain memory domains, like short-term and working memory, may be more affected in breast cancer patients. Consider integrating findings from other studies in the discussion to support your interpretations.
A: Thank you very much for the suggestion. We try to improve the discussion in line with the valuable recommendations [please see pages 15 and 16, lines 300-311].
- the evolving landscape of breast cancer treatment, especially with the introduction of targeted therapies like CDK inhibitors and trastuzumab emtansine, raises an intriguing question about potential impacts on cancer-related cognitive impairment (CRCI). Consider including a section or discussion on their potential impact on cancer-related cognitive impairment (CRCI) compared the chemotherapy. Overall, the effect of treatment type should be discussed more broadly.
A: Thank you for your observation. We try to follow the recommendation. We hope to go in line with expectations [please see page 18, lines 387-395].
- Discuss the practical implications of your findings for breast cancer patients, clinicians, and researchers. How can these findings inform patient care, treatment decisions, or interventions aimed at addressing memory impairments in this population?
A: Thank you for your suggestions. This information has been included. [please see pages 18-19, lines 416-430].
- The research methods chosen for this study are adequate, the results are well-presented and discussed, and the work lies within a topic of high current interest.
A: Thank you very much for your careful review.